# Using Web Video Conferencing to Conduct a Program as a Proposed Model toward Teacher Leadership and Academic Vitality in the Philippines

Janette Biares Torrato [1,*] , Socorro Echevarria Aguja [2] and Maricar Sison Prudente [3]

1 Grade School Department, De La Salle Zobel School, Muntinlupa City 1780, Philippines
2 Graduate School Department, De La Salle Araneta University, Malabon 1475, Philippines; socorro.aguja@dlsau.edu.ph
3 Department of Science Education, Bro Andrew Gonzalez FSC College of Education (BAGCED), De La Salle University, Manila 1004, Philippines; maricar.prudente@dlsu.edu.ph
* Correspondence: torratojb@dlszobel.edu.ph

**Abstract:** The COVID-19 pandemic prompted private basic education in the Philippines to hold a professional developmental program for faculty members using web video conferencing (WVC). Given the uncertainties of WVC educational quality and the challenge of shifting to a fully online environment, this study aimed to evaluate the use of web video conferencing and the development of a faculty development program on sustaining teacher leadership and academic vitality through research. The training was held for nine weeks, and 33 faculty members of the institution participated. Quantitative data include survey questionnaires on perceptions on action research, perceptions on technology integration, training proficiency, and pedagogical practices. Qualitative data include analysis of video recordings, reflection journals, observation notes, and actual research outputs. It utilized descriptive developmental action research using a convergent mixed-method approach, and thematic analysis. Findings show that the use of WVC as a mode of delivery proved successful in creating an effective educational experience for all the participants as evidenced by their enhanced teacher leadership skills and academic vitality. These skills were demonstrated through their willingness to promote professional inquiry as shown by their actual research outputs and demonstration of high proficiency in the adoption of technology integration.

**Keywords:** web video conferencing; online distance learning; teacher training; academic vitality; teacher leadership; action research

## 1. Introduction

Web video conferencing (WVC) has become an essential component in the world of education and many other fields [1]. This has been exponentially magnified in the educational setting due to the pandemic. In this new normal, the use of WVC to deliver instructions and training is widely used as a way of teaching, such that without this platform, teaching would be challenging [2]. This method has given schools new ways of presenting materials, working with teachers and students; thus, they stimulate the development of strategies that are consistent with new technology [3]. WVC uses synchronous two-way audio and two-way compressed video through the Internet. Participants use special cameras, look at monitors and use the microphone of a computer or mobile device at any location. In doing so, they can communicate with each other and with the experts, as if they were in an educational institution. They receive instructions and information on any topic through the platform and can ask questions from all locations involved in the video conference. The use of digital images, videos and video conferencing in the classroom places teaching beyond printed resources and connects participants with the world they live in. WVC, as a form of distance education, well illustrates the relationship between the use of technology

and the need for reorganization to maximize their benefits [1]. Distance education, when properly organized and structured, also illustrates the ability to reach new target groups and to expand the range of educational offerings through technology.

With the advancement of e-learning during the pandemic, the use of WVC has become more widely used in many educational setups. For example, WVC was used to continue students' educational activities by offering courses online. Fatani evaluates student satisfaction with the teaching quality of case-based discussion sessions conducted through WVC using a survey questionnaire on educational quality [4]. Gonzales-Zamora et al. discussed clinical cases and presented diagnoses and treatments to medical students using WVC. This educational strategy has been applied by several teaching sites in a successful manner [2]. Gladović et al. explored the possibility of using web video conferencing and its application in education using free chat and video communication tools [3]. Maximizing the features of a web video conferencing app to establish an active, dynamic, engaging, and educational experience for students and teachers is introduced [1].

WVC has flourished with the development of various services on the Internet. WVC, as a way of open and distance learning, provides education and training in a more flexible way versus regular methods of teaching. Technology is an essential component of this e-learning technology [1–4]. The use of WVC as a mode to deliver professional development programs in this time of the pandemic is essential to ensure continuity of professional growth.

### 1.1. Professional Development Programs

A professional development program for teachers has always been regarded as a major component to enhance their skills and competencies. Investments on improving teachers professionally are of paramount importance [5,6]. Since the 1980s, significant educational initiatives on elevating the quality of teachers and teaching conditions have been made [7]. Some of these are the performance-based compensation system, such as giving merit pay, designing and conducting personal and professional growth, and mentoring other teachers [6,7]. In the Philippines, the passage of Republic Act No.10912, Continuing Professional Development (CPD) Act of 2016, is aligned with the goal to achieve improved performance [8]. The strong position of the government to improve quality of professionals in the country, including teachers, is highly evident as stated in the Declaration Policy of the CPD Law.

This development suggests the core need to develop highly effective professionals, particularly the teacher leaders to promote quality education and consequently, improved educational practices.

According to the literature, a high level of teacher leadership and academic vitality remains to be the goal of many institutions [9,10]. Teachers have a mandate to deliver quality and improved education, while at the same time, to produce knowledge in their fields, integrate technology and the latest research results into their teaching, and conduct research to continue the production and advancement of knowledge [11]. Many institutions seek to develop institutional climates that foster characteristics of teacher productivity to elevate the status of the institution and meet the demands of all stakeholders [9,10,12]. The literature identifies these as the teacher leadership characteristics and academic vitality. The literature further infers that teacher leadership and academic vitality are unique to the type of education institution and its mission and therefore produces a variation in the ideal types of vital teachers [13]. A productive institution is the product of three categories of vital characteristics. These include teacher, institutional, and leadership characteristics. Moreover, leadership and academic vitality within teachers are attributes of institutions that foster vitality in their teachers. Such characteristics include clear and genuine school vision (mission with distinct goal), evident climate and culture of excellence, effective leadership, opportunities for professional growth and development, fair, equitable, and ample rewards and recognition, and sufficient resources. Such institutional characteristics manifest themselves in categories of evidence among teachers. These categories of evi-

dence include professional efficacy through teacher leadership, conduct of research, and proficiency of technology integration [14–17]. These pieces of evidence are construed as drivers supportive of academic vitality. This concept is illustrated in Figure 1.

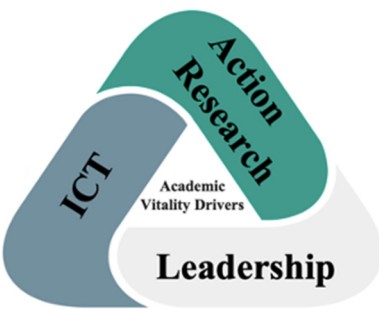

**Figure 1.** Academic vitality drivers.

### 1.2. Teacher Leadership

The principle of teacher leadership is connected to the leadership characteristics of the teacher leaders and contextualized on their teaching and learning environment. Indicatively, the concept of teacher leadership suggests that teachers play crucial roles in the teaching and learning operations of schools [18–20]. If the desire of an organization is for a change to happen, it is important that all members, specifically the teachers of the institutions, in all departments are actively involved. Educational management and development will occur with the leadership of teachers in assessing initiatives and facilitating learning experiences with the end goal of improved student achievement. Based on literature, to train teachers to lead, it is necessary that the core of their learning process is about their own concerns and issues, conducting themselves as learners, reflecting on their practices and sharing their studies with others [21–23]. This makes action research a large component in developing teacher leadership skills.

### 1.3. Action Research

Research is an integral component of any educational institution [24–27]. It provides a powerful framework for drawing upon the knowledge and experiences of teachers to address important educational issues [28–31]. It is an effective professional process that impacts daily and/or future teaching, and the action research process elicits change as it actively engages the teachers with specific issues of concerns [32–34]. When a teacher is engaged in the conduct of action research specifically tailored to the needs and concerns of students, they are empowered to find real and practical solutions required for effective change to occur. According to literature, teachers engaged in the conduct of research most likely become transformative within their professional domain because they are provided with the technical skills and specialized knowledge; hence, they are led to become innovative in their professional lives [35–39].

### 1.4. Technology Integration

With the occurrence of the COVID-19 pandemic, the need to integrate technology in teaching has been exponentially magnified [40]. Technology learning environments framed the educational landscape profoundly [41,42]. Technology plays a crucial role in the delivery of instruction and in accessing the higher-order competencies that are referred to as transversal competencies, namely, critical, and innovative thinking, interpersonal skills, intrapersonal skills, global citizenship, media, and information technology [40,43–45]. These competencies are necessary to thrive and be productive in today's world. Advancements in technology can turn one's life dramatically [46,47]. Hence, the need to redesign and restructure teaching and learning. Teachers are major players in the educational landscape. They are the facilitators of knowledge and skills development to ensure that their students become digitally literate and competent human capital [48–51]. Thus, the integration of

technology into education is a substantial issue for supporting and updating teachers' professional development in today's world [52–61].

However, literature posits several challenges such as many teachers do not regard the conduct of research as part of their trifocal functions, apprehensive to assume leadership roles in promoting professional inquiry and anxious in the adoption of technology integration. With these challenges, it is crucial to develop programs that will empower them to become teacher leaders and improve their academic vitality, thereby enabling them to address key issues on education. Hence, a program focused on these aspects was created and implemented, titled Sustaining Teacher Leadership and Academic Vitality through Research, or STAR program. The design of the program aims to capacitate the teachers in the conduct of action research thereby, enhancing their leadership skills and helping them create innovations, especially in technology integration. Incidentally, due to the occurrence of the pandemic and suspension of face-to-face classes, the use of WVC as a platform to deliver the program was implemented. To be able to evaluate the efficacy of WVC platform, the Community of Inquiry framework was utilized.

### 1.5. Community of Inquiry (CoI) Framework

The community of inquiry (CoI) framework is a social constructivist model of learning processes in online and blended environments. The framework is built upon three dimensions: (1) Teaching presence is defined as the design, facilitation, and direction of cognitive and social processes for the realization of meaningful learning. This involves the (i) instructional design and organization of the course and activities, (ii) facilitation of the course and activities, and (iii) direct instruction. (2) Social presence refers to the ability to perceive others in an online environment as "real" and the projection of oneself as a real person. Social presence involves open communication, affective expression, and group cohesion. (3) Cognitive presence is the extent to which learners can construct and confirm meaning through sustained reflection and discourse. The goal of the community of inquiry is to build a solid foundation of social presence and teaching presence to stimulate cognitive presence in a course [62,63]. Figure 2 illustrates the CoI model.

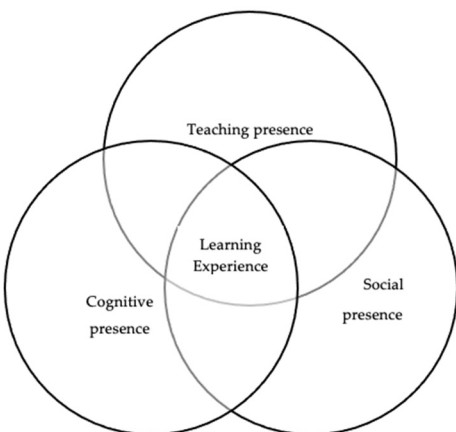

**Figure 2.** Community of inquiry model (Garrison, Anderson, & Archer, 2000).

Teaching presence is the design, facilitation, and direction of cognitive and social processes, while social presence is the ability of the participants to identify with the community, communicate purposefully and develop inter-personal relationships by way of projecting their individual personalities [64,65]. These three interdependent elements are essential for delivering a successful synchronous and asynchronous online course and creating an effective educational experience. According to Moore, using interactive teleconference media offers the opportunity for dynamic inter-learner dialogue and student engagement, thereby enabling teachers to bridge the psychological and communication distance between trainers and participants and among the participants. It also provides the

opportunity to develop the cognitive skills of analysis and autonomy [66–68]. Given the novel exposure to distance education for faculty members and trainers, this study aimed to evaluate the use of the WVC with the training quality of professional development program conducted through WVC. Furthermore, participant feedback about the effect of the training to their own learning experiences will substantially impact future professional development program planning and design as further online trainings are implemented [62,67,69].

Consequently, as established in the discussion above and to address the demand, this study intends to evaluate the use of web video conferencing and the development of a faculty development program on sustaining teacher leadership and academic vitality through research.

### 1.6. Research Problems

This investigation aims to evaluate the use of web video conferencing and the development of a faculty development program based on the key drivers of academic vitality, namely conduct of research, teacher leadership and technology integration. Specifically, it aims to answer the following research questions:

1.  How do the teachers perceive and understand action research in terms of:
    1.1.  the principles of AR;
    1.2.  attitudes toward the conduct of AR;
    1.3.  processes in performing the AR?
2.  Is there a significant difference to the teachers' perception and understanding of action research before and after the training?
3.  How do the teachers perceive technology integration in terms of:
    3.1.  their proficiency on the use of hardware and software;
    3.2.  the factors that influence their adoption and integration of technology in teaching;
    3.3.  their concerns on technology integration in education?
4.  How do the teachers sustain a desired level of research excellence?
5.  What is the effect of the use of WVC on the conduct of STAR program as a model toward teacher leadership and academic vitality?

## 2. Materials and Methods

### 2.1. Research Design

To gain an in-depth understanding of the topic, this study has been carried out using the convergent parallel mixed-methods approach [70–72]. The research process can be symbolized as qualitative and quantitative. A convergent parallel design entails that the researcher concurrently conducts the quantitative and qualitative elements in the same phase of the research process, weighs the methods equally, analyzes the two components independently, and interprets the results together. With the purpose of corroboration and validation, the researcher aims to triangulate the methods by directly comparing the quantitative statistical results and qualitative findings. In the research process, two datasets have been obtained, analyzed separately, and compared. The qualitative data is analyzed using the thematic approach [73]. The research process in this study is given in Figure 3.

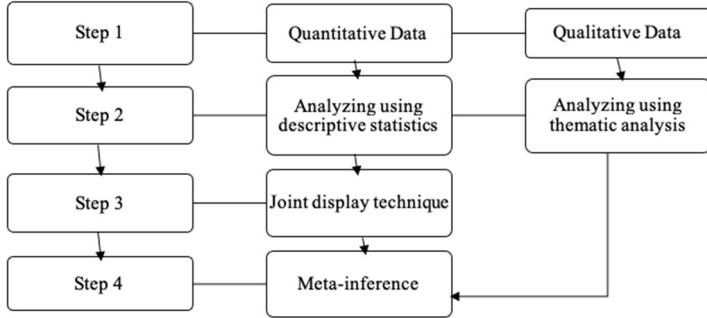

**Figure 3.** Flowchart of the procedure in implementing a convergent mixed method design. Adapted with permission from ref. [70]. Copyright 2011 Creswell & Plano Clark.

### 2.2. Research Participants

A total of 33 faculty and administrators from a private basic education school in the Philippines initially and voluntarily participated in a three module-series webinar workshop named as Sustaining Teacher Leadership and Academic Vitality through Research (STAR) Program. The group was composed of top management administrators, 6% (i.e., director and vice principals); middle administrators, 33% (i.e., subject area coordinators, strategic planning officer); subject teachers; academic services faculty, 61% (i.e., subject teachers, formators guidance counselor). Inclusion criteria were that participants must be members and employed in the institution where the study was conducted, willing to be part of the study, and complete the extensive 9-week webinar workshop.

### 2.3. Research Instruments

For the quantitative data, the following research instruments were used.

Principles on Action Research Questionnaire (PARQ). This 30-item survey questionnaire was developed by Prudente and Aguja [74]. The items were drawn from the themes emerging in the literature on action research. This questionnaire was validated based on the Philippine context. All the items on the questionnaire required the participant to respond to a four-point Likert scale: 1—strongly disagree, 2—disagree, 3—agree, and 4—strongly agree.

Perceptions on Technology Integration and Training Proficiency and Pedagogical Practices Questionnaire (PTITPPP). This questionnaire aims to measure the technology proficiency and perceptions of teachers on pedagogical practices [75]. There are three sections in the survey questionnaire: proficiency on the use of hardware and software; factors affecting adoption of technology integration; attitude on technology integration. These sections contain 39, 29, and 11 items, respectively. Proficiency on the use of hardware and software and factors affecting adoption of technology both include a 5-point rating scale with options ranging from "(1) Not Proficient to (5) Proficient". Attitude on technology integration include a 5-point rating scale with options ranging from "(1) Do Not Agree to (5) Agree". Proficiency on the use of hardware includes knowledge and skills of teachers on the use of desktop, laptop, external hard drive to name several; software refers to use of school email system, web browsers, conducting online searches, creating e-portfolio, etc. Factors affecting adoption of technology includes integrating technology gives teachers greater control over instructional performance, improving instructional performance among others. Attitude on technology integration pertains to the teacher being concerned on students' attitude toward technology and helping other teachers in integrating technology.

For the qualitative data, instruments used were reflection journals, program evaluation, action plan, action matrix, and abstract manuscript, among others. All instruments were coded following the coding in Table 1.

**Table 1.** Coding of Instruments.

| Quantitative Data | Codes |
|---|---|
| Principles on Action Research Questionnaire | PARQ |
| Perceptions on Technology Integration and Training Proficiency and Pedagogical Practices Questionnaire | PTITPPP |
| Qualitative Data | |
| Reflection Journal | RJ |
| Sustaining Teacher Leadership and Academic Vitality through Research | STAR |
| STAR Program Evaluation | SPE |
| STAR Program Impression | SPI |
| Action Research Action Plan | ARAP |
| Action Research Matrix | ARMx |
| Action Research Abstract | ARA |
| Action Research Manuscript | ARMN |
| Google Chat Feedback | GCF |
| Gmail Feedback | GMF |
| Researcher Observation Notes | RON |

*2.4. Research Ethics*

Ethical considerations were given throughout the study regarding access, confidentiality, and consent. Prior to the conduct of the study, permission from the school president was obtained. Upon approval of request, the details of the STAR program were shared with the HR director and academic heads for information and request for support. The letter to the employees was drafted and shared to the HRMDD Director for comments. After it was approved, the email was sent to all faculty and administrators. Interaction with participants were performed after prior appointment and approval from them. The first email contained an invitation to attend the webinar series on the conduct of educational action research. Initially, 50 faculty and administrators signed up. A follow-up email containing the schedule, purpose and objective of the program, profile of resource speakers, expectations, and information that the program was part of the dissertation was sent to the fifty volunteers. They were asked to acknowledge the email to confer agreement to participate. Among the fifty volunteers, thirty-three confirmed their attendance and willingness to participate in the study. Some declined after learning that the sessions will last for nine (9) Saturdays, stating that they cannot commit to be present in the duration of the webinar series, while the others merely did not reply after the second email.

The research was conducted in a manner that was respectful to the participants and everyone who was influenced by the research process.

*2.5. Research Procedure*

Figure 4 illustrates the process in conducting the webinar workshop [76]. An explanation of the process follows.

The STAR program was conducted through synchronous and asynchronous sessions. Synchronous sessions refer to online lecture and discussion using teleconference. Asynchronous sessions refer to engagements outside the teleconference. Experts on the conduct of educational action research were invited to facilitate the webinar workshop. They are known both in local and international arenas, are trainers of teachers and supervisors in basic and tertiary levels both in private and public schools, are research authors and widely cited as researchers in reputable research journals, are much sought speakers in national and international conferences, and they were specifically chosen to ensure that the

participants are trained by credible and well-respected consultants in the field of research. The researcher served as the moderator, observer, and a conduit between the consultants and participants. She prepared the logistics and all other requirements for the STAR program. Materials included action research modules, reflection journals, researcher-made evaluation tools, and survey questionnaires.

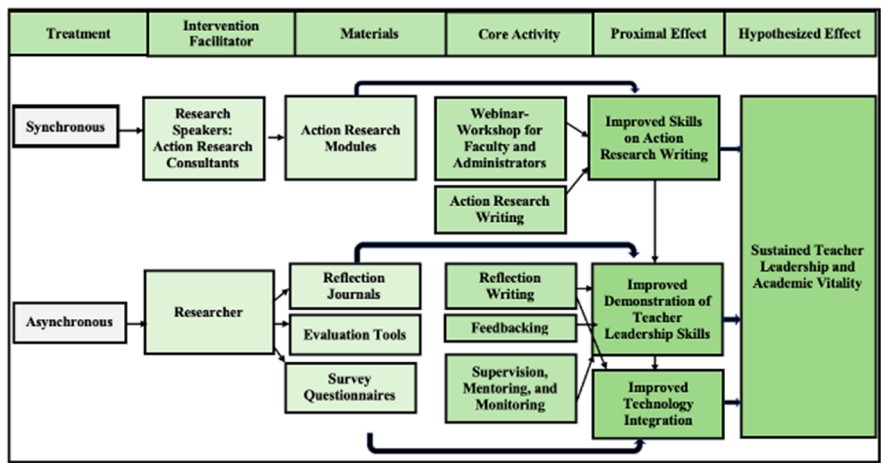

**Figure 4.** The STAR program boost logic model. Adapted with permission from ref. [76]. Copyright 2019 SAGE.

The goal of the webinar workshop was to improve the research skills writing of the participants. To achieve this, the facilitators provided workshops to volunteer participants over the course of one term. The sessions covered skills that have been shown to be causally related to success in research writing. The modules specified nine 3 h synchronous sessions, including writing relevant action research papers related to their field of interest and current areas of specialization. Each session followed roughly the same set of activities: introduction to the session topic, workshop, critiquing, question, and answer. Webinar sessions occurred at a regular interval throughout the term, providing participants a week between sessions to use their new knowledge in their classroom and offices.

The resource speakers or consultants designed the webinar to adapt to the needs and context of the participants. First, by helping them acquire and incorporate new skills in identifying areas for improvements in their current practices through developing action research proposals; second, by collecting and analyzing evidence of improvement using qualitative and quantitative data; third, by disseminating the action research results.

The sessions involved active learning, maintained coherence in format and content from session to session, provided enough time for faculty and administrators to learn about and practice the skills and concepts covered by the training, and required participant's collective participation. All these characteristics are core features of high-quality teacher professional development programs [76].

*2.6. Data Collection*

2.6.1. Quantitative Phase

All survey instruments were administered online via Google Forms. The pre-survey for PARQ was administered before Module 1 Session 1 started. The post-survey PARQ was administered after the session on Module 3 Day 2. The PTITPPP was administered on Module 1 Session 2 and 3, respectively. Response rate was 94–98% for the 33 participants.

2.6.2. Qualitative Phase

The data was collected from multiple sources and included: (1) virtual discussions; (2) participants' responses to the reflection journals and program evaluation forms; (3) actual action research outputs; (4) feedback; and (5) observation notes.

### 2.7. Data Analysis

2.7.1. Quantitative Data

Reliability and validity of the survey questionnaires were established using descriptive statistics, frequency distribution, correlation, and effect size. Data were entered in SPSS software. Tables 2 and 3 show the description and range for the responses on survey questionnaires on PARQ and PTITPPP.

**Table 2.** Description and range for survey responses for PARQ and PTITPPP factors and concerns.

| Likert Scale | Interval | Response Description |
| --- | --- | --- |
| 4 | 3.50 to 4.00 | Strongly Agree (SA) |
| 3 | 2.50 to 3.49 | Agree (A) |
| 2 | 1.50 to 2.49 | Disagree (D) |
| 1 | 1.00 to 1.49 | Strongly Disagree (SD) |

**Table 3.** Description and range for survey responses for PTITPPP software and hardware.

| Likert Scale | Interval | Response Description |
| --- | --- | --- |
| 5 | 4.00 to 5.00 | Proficient |
| 4 | 3.00 to 3.99 | Somewhat Proficient |
| 3 | 2.00 to 2.99 | Uncertain |
| 2 | 1.50 to 2.00 | Somewhat Not Proficient |
| 1 | 1.00 to 1.49 | Not Proficient |

2.7.2. Qualitative Data

The data were examined through thematic analysis using the coding system in accordance with conceptions obtained from data [73]. In this method, the important dimensions were determined in relation with the purposes of the research. Codes were created depending on the emerging meaning with reference to the data. The codes are associated, and themes were defined. Data were entered in MAXQDA, a software program for qualitative data analysis.

Specifically, the deductive way was used in this study. This approach directs coding and theme development to existing concepts or ideas. The themes were summarized in accordance with the constructs in the conduct of action research, teacher leadership readiness, and technology integration.

Figure 5 represents the process.

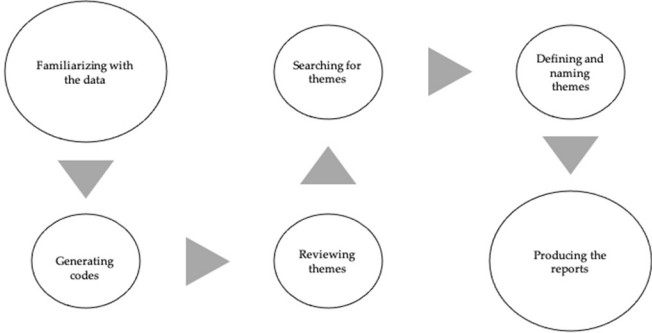

**Figure 5.** Phases of thematic analyses. Reprinted with permission from ref. [73]. Copyright 2006 Clarke, Braun, Terry & Hayfield.

Credibility and validity of the findings were secured by triangulating different sources of information, respondent validation, adequate engagement in data collection, critical

self-reflection or reflexibility, audit trail, and rich thick description [71,72,77]. To further establish the validity of the findings, informal external validation was conducted.

Relatedly, after generating the quantitative and qualitative data results, the meta-inference was also identified. This is described as "meta inference as an overall conclusion, explanation or understanding developed through and integration of the inferences obtained from the qualitative and quantitative strands of a mixed method study" [78]. They describe the importance of an "integrative framework" for making validity claims for mixed-method research. The integrative framework seeks to distinguish between *inference quality* (an attribute of the process of meaning making and/or its outcomes) and *data quality* (attribute of the inputs to the process of meaning making). According to the literature, there are four types of meta-inferences. These are converging, complementary, expansion and discordance. Converging inference means that both results support the idea; complementary means there are different ideas but not conflicting; expansion provides a central overlapping, broader interpretation, or hybrid of complementary; and discordance means ideas are different and conflicting [79].

## 3. Results

The narrative below is a detailed description of the activities that transpired during the webinar series from Sessions 1–9. Figure 6 shows the timeline of the workshop.

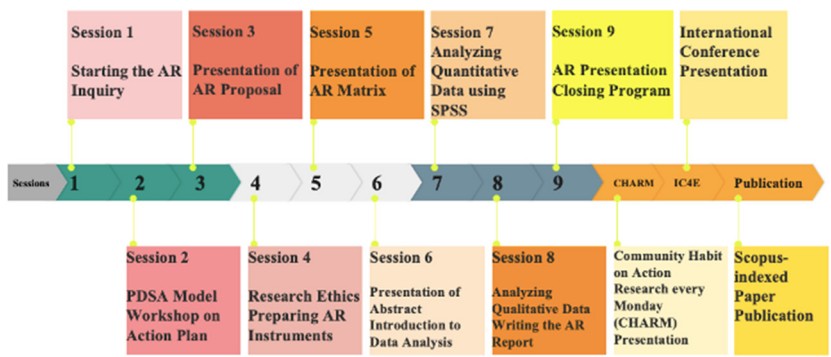

**Figure 6.** STAR Webinar Workshop Timeline.

### 3.1. Session 1

In Module 1 Session 1, prior to the main activity, participants were asked to answer the pretest on Perception on Action Research Questions (PARQ). Then, prayer and introduction of participants and resource speakers followed. A short overview of the program was shared with expectations. After which, the speakers delivered the first topic. This was about the principles and processes in the conduct of action research. After discussing, the participants were asked to answer reflection questions about a topic that they would like to study. Four questions were shared, and they were given 10 min to write down their ideas. All participants showed interest and enthusiasm. All of them were able to answer the reflection questions using Google Docs. The remaining time was used for presentation and critiquing of initial plans. Six of them were able to share their outputs. After the session, the resource speakers were impressed with the enthusiasm and output of the group. They said that participants came ready because they were able to write their concepts about their planned action research. At the end of the session, the participants shared feedback via the Google Hangouts/Chat.

> *"Thank you for the beautiful and productive day. Happy to be part of the team."*

> *"Thank you for organizing this AR seminar. It is part of your dissertation, but you are able to help many."*

Below are some of the answers in Reflection Journal.

*"Contrary to what our good professors mentioned, I didn't come prepared. But the lecture this morning helped distill my thoughts and prompted me to focus on a possible topic which, for me, is a pressing need in school. The comments were helpful in putting more structure and direction to my proposed AR. Excited to work on this and looking forward to more productive Saturdays in the coming weeks!"*

*"It is indeed a informative session in which theory and practice are dynamically fused resulting in a paradigmatic understanding of everything discussed and shared in this academic session. My sincerest gratitude to all, our resource persons, organizers and fellow educators for an amazing sharing of knowledge."*

*"Action research is #careergoals. It reflects one's teaching practices. It is a self-evaluation of one's role to spark good change. Simply, it is about me."*

*"Thank you for this opportunity. The session was informative and helpful especially to teachers who are into research writing."*

*"Our session today is not only helpful to me as a research enthusiast, but also as a research teacher myself. I find the introduction (characteristics, principles of AR, etc.) helpful to my own discussion of AR next term. Hopefully, I may be given permission to adapt the slides so I could share what I have acquired to my students in the NHS."*

*"This first meeting provided informative guide in creating a research proposal. The 5-step guide really helped me in eliminating the distractions in planning a research proposal. This helped me come up with an action research that I am happy to share to the community once completed"*

### 3.2. Session 2

The topic was about the plan, do, study, act (PDSA) model. The participants were introduced to the model to follow in conducting action research. The first part was dedicated for the lecture while the second part was used for workshop. The participants were asked to accomplish an action plan in 10 min. Five were able to present for critiquing. Before the end of the session, the participants were asked to submit an accomplished action plan before the next session the following week. They were advised to submit the output to the researcher, who also served as the moderator on Thursday afternoon, which in turn will be submitted to the resource speakers on the same day. Outputs with comments were returned on Friday in preparation for Session 3. At the end of the session, the participants were asked to create a hashtag to sum up their Session 2 experience using Padlet.

The week after Session 2, consent letters to parents and employees were released. The exchange between the participants and researcher was constant. Several of them asked for advice about data management, authorship, protocol in adapting tools from other authors. Some merely expressed gratitude for being given the opportunity to attend such a webinar. Most participants were expressive of their appreciation to the knowledge and skills of the resource speakers.

### 3.3. Session 3

Session 3 was dedicated for the proposal presentation. Sixteen participants were able to present their work for critiquing while eleven were finished in Sessions 1 and 2. The presenters were ready. Comments on the presentations were mostly focused on making the study explicit and focused. Detailed comments included making the study specific and not too general, defining terms specific to the study, and preparing indicators or manifestations of learning, motivation, engagement. As speakers gave feedback, the others were writing notes and making revisions on their outputs; thus, learning was occurring as each participant reported. It was evident that the level of engagement was high as indicated by their active participation and answers to the reflection journal. After the session, participants were still engaged. Some expressed appreciation; others asked advice. The participants were highly motivated and engaged.

### 3.4. Session 4

This was the start of Module 2. There were 27 participants who attended the session. Topics were centered on protocols on conducting an interview and research ethics. Some questions raised were about the number of participants in an interview, clarifications on focus group discussion, ethical concerns, and data ownership.

After the first four sessions, the researcher was able to obtain a better profile of the group as evidenced by their actions, sharing, ideas, inputs, and questions. Below is the observation of the researcher.

> *"Participants of STAR 1 were volunteers. They enlisted on their own despite knowing the rudiments of the workshop. They know it will be held in nine (9) Saturdays with three (3) hours per session. They were fully aware that engagement in AR is an additional task. However, they were ready. They were excited with the tasks ahead of them. Perhaps, because they were intrinsically motivated, highly engaged to grow professionally, and most of all—research enthusiasts."*

### 3.5. Session 5

This was another meaningful and productive session. It started with a prayer, followed by presentation and critiquing of action research matrices. Normally, participants submit their work to the researcher every Thursday afternoon. The researcher emails their work to the experts, and then outputs with comments are given back to the researcher, who in turn will share it with the participants. Once received, participants revise the outputs based on the suggestions. These are presented in the next session. Participants were given five minutes each to present the research questions, data needed, data gathering tools, and data analysis.

On this day, 17 participants were present. Some presenters shared near-perfect outputs, while others were advised to modify either the questions, tools and/or data analysis. After the presentation, the other participants shared that they were writing notes and making modifications to their own presentations. Normally, it takes 17 or 18 participants to present in three hours.

It was evident that as the sessions progressed, the proposed action research studies of the participants were becoming clearer and better. The intent of their studies became more focused. The outputs showed better understanding of the participants' idea on the conduct of action research. Some of them were praised for almost perfect outputs. The manifestation of learning was highly evident.

At this time, it was already evident who among the 33 participants were truly engaged. Sixty percent of the participants were consistent in submitting all required outputs. Several participants attended but did not submit outputs. Some reasons for not attending the synchronous session were personal concerns related to health, going to the doctor, attending to family members, attendance to other work-related webinars, and graduate school, among others. Survey questionnaires and reflection questions were shared every session; however, the turnover was not always 100%. For the survey questionnaires, the turnout was at 100% while the reflection was at 80%. Despite the follow up, the researcher was not able to collect 100% submission. Nonetheless, the results of the survey questionnaires and reflections proved to be aligned with the discussion, class interactions, feedback, actual research outputs and observations.

### 3.6. Session 6

This session was dedicated to the presentation of abstract. Fifteen participants were able to submit prior to the session on that Saturday and were given comments. On the day itself, four also submitted. There were 18 who presented their studies for critiquing. Based on the outputs of the presenters, the studies were becoming more clear. Some recommendations were focused on the revision of the title by making it more focused, including specific action and how to analyze the data (quantitative or qualitative using thematic analysis). On this day, eight participants were absent. Three (3) were attending

graduate school, one (1) had to accompany his mother to chemotherapy, one (1) was pregnant and needed to visit her OB, three (3) did not provide any information.

Directly after the session, the researcher made a follow-up to those who did not attend. One appreciated the effort for asking. He asked for a consultation time. This was performed in the afternoon after Session 6. Some replied that they have problems with connections. Another one replied that she could no longer attend due to graduate school but requested if she could stay in the Google classroom because she was learning a lot.

This time, the tasks were becoming more technical and difficult. Hence, the researcher offered help by providing consultation times twice a week. Those who needed help or had any concern could request for a meeting using the GMeet Link. One immediately requested a consultation time. Another requested for validation of survey questions. Some participants that were followed up replied and signified interest in making up unperformed tasks. The rest acknowledged and appreciated the effort.

### 3.7. Session 7

This session was scheduled for a lecture on data analysis using SPSS. Some participants shared raw data. The resource speakers showed how to navigate and use the SPSS software to analyze pretest and post-test, obtain frequency distribution, compare means and correlations, and create graphs, among others.

### 3.8. Session 8

Sample data using SPSS were shared by the participants. The resource speakers gave additional recommendations on how the participants could further analyze the data. This time, the topics were more technical. There were six (6) consistent participants who were making progress each session. They were able to collect and analyze the data using the SPSS.

In addition, two other topics were also included in this session: "How to write the AR report and using qualitative data to analyze". This was created to prepare the participants to write their own manuscript.

### 3.9. Session 9

This was the last day of the session. This was a celebration of success. Among the 32 participants, there were six who consistently showed progress. They were able to complete the action research within the duration of the webinar workshop. They were chosen to present their studies to the group. This session was also the closing program. The top administrators of the institution were invited to witness the celebration.

During the research presentation and the entire program, feedback from the participants were overwhelming. Below are some of them:

> "To all the presenters, my sincerest CONGRATULATIONS! I feel so happy and inspired by the quality of action research work that you are all doing!"

> "Kudos for her leadership in organizing the STAR...Animo!"

> "Thank you, Sir! You have all inspired us to continue infecting others with the AR virus! Animo!"

> "Thank you so much for the opportunity to be trained by research scholars. In our 9-week webinar, I became a fan! I cannot thank you enough for motivating us to be better educators. Thank you very much! I am working towards finishing my paper soon."

> "Thank you for sharing your insights with us. Teachers like you inspire students and your colleagues and more so to us AR facilitators. Let us continue to empower each other and ensure quality education in our institution!"

> "Thanks for the good realizations, and appreciations re the conduct of the AR webinar workshop...and the nice words uttered!"

*"This is the virus we want to have. We're grateful for your wisdom and expertise! Looking forward for more! God bless your heart always. Cheers and blessings !"*

*"Thank you so much for the gift of the support group that helps each other to shine and spark light to others as well. Thank you for being my constant star when I am lost in my journey towards the completion of my AR. Your constant guidance makes the journey meaningful and easy. It is a blessing to work with you and it's a privilege to have been mentored by you. I am humbled and honored of how you dedicated your precious time with us despite your busy schedules. Know that this journey will not end with my presentation here. I want to pay it forward to affect the AR virus to all educators. To my fellow participants, congratulations."*

*"Our sincerest thanks for your passion, dedication, and boundless patience in leading us towards this program's culmination. You are not done with is yet, that we can assure you as we intend to put up the Research Commons which we hope you will take on as advisers."*

After the last session, the six who were able to finish the AR continued their journey. They presented in local and international conferences, where three of them were chosen as best presenters. Their manuscripts were also accepted for publication in Scopus-indexed proceedings.

With the detailed description of the webinar, below is the discussion to the answers for each research question.

*3.10. Research Question 1*

1.   How do the teachers perceive and understand action research in terms of:

    1.1.   the processes of AR;
    1.2.   attitudes toward the conduct of AR;
    1.3.   principles in performing the AR?

Table 4 shows the joint display of mixed data on the process on the conduct of action research both for quantitative and qualitative data. The integration involves merging the results to make better comparisons and a more complete understanding.

**Table 4.** Joint display of mixed data on teachers' perception and understanding on the conduct of action research.

| Quantitative Data. | | | | | Qualitative Data | Meta-Inference |
|---|---|---|---|---|---|---|
| Construct | Survey | M | SD | Cohen's-d | | |
| Process | Pre Survey | 3.16 | 0.16 | | Clear Action Plan | Complementary |
| | Post Survey | 3.25 | 0.21 | | | |
| Attitude | Pre Survey | 3.11 | 0.35 | d = 0.7206 | Passion and Productivity | Expansion |
| | Post Survey | 3.34 | 0.31 | The effect size is from medium to large. | | |
| Principles | Pre Survey | 3.52 | 0.39 | | Improvement Theory | Expansion |
| | Post Survey | 3.69 | 0.31 | | | |
| General Average | Pre Survey | 3.25 | 0.25 | | Regular Program | Complementary |
| | Post Survey | 3.41 | 0.19 | | | |

Table 4 shows that based on the means, the participants in general had positive perception and understanding toward the processes, attitudes, and principles of action research. The participants agreed with the processes and attitudes in performing action research and strongly agreed with the principles of action research. The size of the differences of the means for the pre- and post-surveys was from medium to large, indicating that there is a significant difference between them.

Based on the integration of the evidence from the quantitative and qualitative data, results show that participant perceptions and understanding of processes of action research were either complementary or expansion.

To illustrate the understanding of the participants further, the qualitative data were coded following thematic analysis to determine the emerging themes in terms of the three constructs. Tables 5–7 show the generated themes from reflection journals, STAR program evaluation, and researcher's observation notes.

**Table 5.** Themes from the varied data sources on the processes in the conduct of action research.

| Code | Category | Theme Description | Generated Themes |
|---|---|---|---|
| Researcher's Practices | Articulates reflection | The action research is reflective of the researcher's teaching practices | Self-reflection |
| Research Focus | Root cause of the problem | The questions are explicit and directed to the intended outcome of the study | Clear Action Plan |
| Assessment of Current Situation | | Refers to the consistency of the research questions, instruments, research design, and data analysis. | |
| Continuous Improvement | Iterative Process | Action research is a continuous improvement that aims to change and improve practices. | Cyclical |
| Address the Challenges | Innovation | Action research has proposed intervention, solution or program that intends to address existing issues, challenges in the classroom setting using varied techniques or tools. | Intervention |

**Table 6.** Themes from the varied data sources on the attitude on the conduct of action research.

| Code | Category | Theme Description | Generated Themes |
|---|---|---|---|
| Work of love | Enjoy trying out new things | It takes great passion, determination, and love to be able to engage in the conduct of action research. | Passion |
| Challenging yet rewarding | | It is a challenging endeavor but fulfilling and rewarding. | |
| Tedious but gives clarity | Effective Teacher | The conduct of action research is taxing but it's gives clarity to the purpose of the lesson and/or teaching | Efficacy |
| Stimulating and productive | Professional Knowledge maker | Action research stimulates the researcher's ideas to become productive in the profession. | Productivity |
| Self-exploration and growth | | Action research is a process that involves self-exploration and professional growth. | Professional Growth |
| Learning Experience | Provision of Training | Opportunity to learn from the experts is a gratifying experience | Opportunity |

**Table 7.** Themes from the varied data sources on the principles in the conduct of action research.

| Code | Category | Theme Description | Generated Themes |
|---|---|---|---|
| Continuous Improvement | Change practices to bring out results | Improvement of research skills and educational practices | Improvement Theory |
| Theory to Practice | Understand the importance of action research in addressing concerns, issues, and problems in school | Fusion of theory and practice to address education issues | Improved Professional Practice |
| Inform Change | Action research is an effective way to evaluate teaching strategies and modalities used in the teaching-learning process | Evaluation of teaching strategies to contribute to the knowledge base | Evaluation of Practices |
| Within the Context of the Teacher | It is an opportunity to further improve the teaching-learning process using the lens of the teacher | Gauge effectiveness of interventions or innovations within the context of the teacher | Teacher Efficacy |

The themes generated from qualitative data sources reinforces that the participants have strong and grounded understanding of the processes involved in the conduct of action research. Although based on the researcher's observation, the processes were a bit ambiguous during the earlier sessions as evidenced by the field note below.

*"The teacher-researchers have vague or profound ideas. The intended improvements of the studies were not clear. They need to determine what exactly do they want to happen, what they can do. Explain the interventions well. The resource speakers reiterated that psychological constructs like motivation and engagement cannot be measured in two weeks. At least, this should be four weeks. Cognitive constructs like achievement, understanding is doable or measurable in one or two weeks although this can only pertain to certain competencies." RON*

This understanding, however, gradually changed from session to session when topics were discussed and explained to the participants. The actual output shows that participants were able to have a solid understanding of the processes at the end of the webinar workshop. This is also evident with the feedback shared by one participant.

*"The processes involved in action research may be tedious, but they help you clarify the direction you need to take toward the completion of your study. The mapping out of your action plan, the writing of research statements, the writing of the research title, abstract, and key words allow you, as researcher, to organize your thoughts and help you focus on what to prioritize, to retrace your steps, and rethink about your plan." (RJ)*

Collectively, the results of the survey questionnaire and other data sources explicate the theoretical underpinnings related to the processes on the conduct of action research [31,33,35,74,80].

Based on the PDSA model of action research, the participants had high perceptions on the processes; however, actual output and reflections during the fourth session revealed their level of understanding on the processes, particularly on the following constructs: Triangulation of data, experiencing, requiring, examining, alignment of AR components must be reinforced.

It was also evident that the habits of highly effective teams, namely, collective inquiry, relentless questions about the status quo, action orientation, turns learning into action, commitment, bring present reality closer to ideal future, results orientation, and improvement being observable and measurable, were demonstrated by the participants in the actual interactions and outputs. This shows that the results of the PARQ survey questionnaire and the qualitative data were aligned.

The second cluster involves attitude. Table 6 shows the attitudes of the respondents on the conduct of action research.

The observation notes and the answers to the reflection questions suggest that the participants have a positive attitude on the conduct of research. It affirms the results of the survey questionnaire, particularly on enjoyment in trying new things, improvement of one's practice and finding joy in trying new things.

Below are notable observations of the researcher on the attitude of the participants.

*"They were volunteers despite knowing that STAR will be an additional load on their end. They know it will be held on nine (9) Saturdays with three (3) hours per session. They were fully aware that they might be going to work hard, and this will be an added task. However, they seemed to be ready. The group seemed to be excited with the tasks ahead of them. Perhaps because they were really research enthusiasts. As the sessions progressed, the participants were engaged. Exchanges in Hangouts were constant about what, how to do things. Participants shared resources, ideas, asked for advice, shared their joys like when a tool was approved by certain authors. Every time the session ends, the researcher received affirmations and gratitude notes."*

Given these observations, it can be deduced that the participants demonstrate positive attitude to learning how to conduct action research. Based on studies, attitude is an evaluative and affective attribution/disposition regarding favor or disfavor of a particular action, situation or subject [81,82]. The linkage between psychological tendencies and behavior has been investigated in large body of research [81–84]. The participants demonstrated high sense of motivation, engagement, and determination. All these lead to a positive attitude while conducting action research.

A similar process was followed to determine the themes for the principles.

Table 7 results are consistent with the responses of the teachers to the reflection logs and open-ended questions. Collectively, the results of varied data solidify the theoretical underpinnings related to the principles on the conduct of action research.

*3.11. Research Question 2*

2. Is there a significant difference to the teachers' perception and understanding of action research before and after the training?

Table 8 provides correlations between the pre-and post-survey results per cluster. The correlations (r = 0.054, 0.026, 0.299 and 0.106) and significance level (0.764, 0.884, 0.092 and 0.557) show that there is no significant relationship between the responses of the participants in pre- and post-survey.

**Table 8.** Paired samples correlations.

| CLUSTER | | Mean | Std. Deviation | Std. Error Mean | 95% Confidence Interval of the Difference | | t | df | Sig. (2-Tailed) |
|---|---|---|---|---|---|---|---|---|---|
| | | | | | Lower | Upper | | | |
| PROCESS | Before Training After Training | −0.08182 | 0.25886 | 0.04506 | −0.17361 | 0.00997 | −1.816 | 32 | 0.079 |
| ATTITUDE | Before Training After Training | −0.23030 | 0.46333 | 0.08066 | −0.39459 | −0.06601 | −2.855 | 32 | 0.007 |
| PRINCIPLES | Before Training After Training | −0.17515 | 0.41750 | 0.07268 | −0.32319 | −0.02711 | −2.410 | 32 | 0.022 |
| GENERAL AVERAGE | Before Training After Training | −0.16030 | 0.29757 | 0.05180 | −0.26582 | −0.05479 | −3.095 | 32 | 0.004 |

The table above shows that at $\alpha = 0.05$, there is a significant difference in the perception and understanding of teachers toward the attitude and principles of action research before

and after the training, while no significant difference is observed in the process of action research. However, in general, there is a significant difference in the perception and understanding of the participants toward action research.

In relation to the CoI Framework, the processes and the results evidently show that there is a solid foundation of teaching and social presence resulting in the full stimulation of the participant's cognitive presence about the goal of the program-related action research [62,63]. The positive feedback of the participants about the expertise of the consultants, the engaging interactions between and among the consultants and participants, and the improvement of the participants skills to conduct research from Session 1 to 9 validates this observation.

### 3.12. Research Question 3

3.    How do the teachers perceive technology integration in terms of:
    3.1.    their proficiency on the use of hardware and software,
    3.2.    the factors that influence their adoption and integration of technology in teaching,
    3.3.    their concerns on technology integration in education?

Table 9 shows that the participants are proficient in the use of technology hardware and software. This result suggests that all participants are proficient in the use of hardware and software in technology devices. Perhaps this is attributed to the technology program of the institution.

**Table 9.** Joint display of mixed data on teachers' perception on technology integration and training proficiency.

| Quantitative Data | | | | Qualitative Findings | Meta-Inference |
|---|---|---|---|---|---|
| Construct | M | SD | I | **Proficient in Hardware and Software** "The teachers can easily navigate the different technology tools. They do not have queries on the technology apps. All throughout the webinar workshop, they were focused on the content. This is indicative that software and hardware are not major issues." | **Expansion** The observation of the researcher on how the participants easily use the devices and navigate the technology applications during the workshop solidifies the positive perceptions on the use of hardware and software as seen in the mean score. |
| Hardware | 4.43 | 0.81 | P | | |
| Software | 4.27 | 0.99 | P | | |
| Construct | M | SD | I | **Teaching Strategies** "Technology allows self-paced learning, gamified activities, and asynchronous access to educational materials. Engagement is highly addressed, and self-evaluation is maintained and enforced." "I believe it can become beneficial in improving my approach and techniques in teaching more effectively to my students and preparing efficiently my lesson." | **Expansion** The participants reflections on how technology affects their teaching and students' learning confirms their adoption and concerns on technology integration as evidenced by the mean scores. |
| Adoption of Technology | 3.51 | 0.83 | SA | | |
| Concerns/ Attitude | 3.23 | 0.93 | A | | |

The institution where the study was conducted has been a front liner in the delivery of the continuity of learning to its students by looking at various scenarios and solutions involving educational technology [84]. With the implementation of the PEARL program in AY 2012–2013 and the transition to next generation blended learning (NxGBL) thereafter, the school has had seven (7) years of a head start with online classes. The school has been able to pivot to a blended model given online learning tools and the proper professional

development of its faculty. It has a wide experience on the delivery of NxGBL [85]. Hence, it can be deduced that the proficiency of the participants can be attributed to the technology program of the school.

The framework in Figure 7 can further explain the reason for the proficiency level of the teachers. It represents the levels of blended learning in the institution.

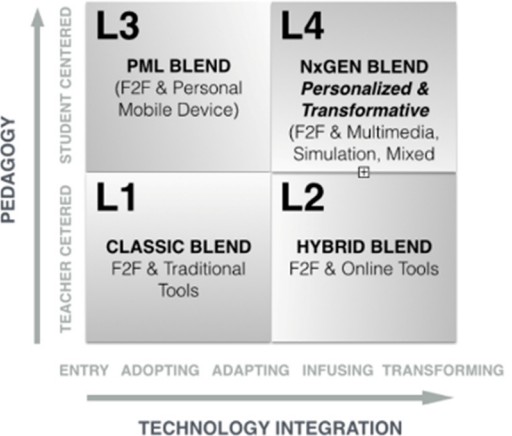

**Figure 7.** DLSZ levels of blended learning derived from NxGBL program.

The results of the survey questionnaire affirm the success of the school's strong emphasis on providing a technology learning environment as espoused in the NxGBL program. Since the school follows a curriculum framework that involves the use of digital learning environment, it can be surmised that teachers are encouraged and required to be proficient in the use of hardware and software [84–86]. Teachers' sustained use of technology resources increases their beliefs and skills on the relevance of learning the technical dimension of technology [87–89]. Digital literacy is a prerequisite to the success of technology integration instruction. Digital literacy drives successful pedagogical practices.

The participants strongly agreed in terms of adoption of technology integration and agreed on the concerns and attitude toward technology integration.

To further validate the perceptions of participants on technology integration proficiency, their actual use of technology tools during the webinar workshop itself, research topics and tools needed to conduct the study and answers to evaluation forms were explicitly observed. Below are notable observations.

Due to the pandemic, technology was used extensively to communicate, collaborate, organize, and engage. Participants presented and asked for feedback using different technology tools and applications: for official communication, the school email system, which is Gmail, was used to create and send; participants used web browsers to conduct online searches; MS Word and Google Docs were utilized to create tables and documents; Spreadsheet or Excel were used to create graphs and share raw data; Slides and/or PowerPoint were used for presentations; participants transferred information from one device to another (i.e., mobile phone to laptop or Mac); the learning management system was Google Classroom; use of hypertext linking was also performed; storing data in Google Drive, using Hangouts for easy and fast communication; Google Forms for survey questionnaire, Google Meet for the teleconference, Padlet for reflections. It was evident that participants were proficient in the use of software and hardware. Most likely, the proficiency of the participants is attributed to the comprehensive technology program of the school, next generation blended learning (NxGBL) program, continuity of learning (CLP). Technology certification is strongly supported by the institution. All faculty members and administrators are highly encouraged to obtain certifications. Meaningful rewards are given for teachers with certifications. Consequently, teachers who obtain them become proficient in the use of hardware and software.

To further explicate the factors affecting them to integrate technology inclusive of their concerns, the abstracts were carefully analyzed. Topics are listed in Table 10 about their main concerns.

**Table 10.** List of Topics.

- AEP Program Evaluation
- Flipped Classroom Strategy
- Students' Engagement in an Online Distance Learning
- Audio Podcasts as Delivery of Formation Content
- Career Guidance Program Evaluation
- Designing Authentic Assessment
- Performance and Engagement using Learning Playlist
- Efficacy of Online Formation Program
- Use of social media
- Multiple Attempts Format
- Digital Technology Acceptance using Learning Playlist
- Robotics Program
- Differentiated Instruction in PE
- Increasing Student Engagement using Playlist
- Using Gamification to Improve Self-determination
- Using Active Learning Online Activities
- Flipped Classroom Learning in ODL
- Supportive Environment for Student Athletes
- Online Synchronous Routine

The list shows participants' concern with technology is focused on strategies, assessments, modalities, technology acceptance. Most likely, these concerns stemmed from the participants desire to deliver quality and effective instruction. These concerns may have been influenced by the technology program of the school and the current pandemic. Most likely, the present and new norm of teaching, online distance learning, may be the greatest factor for them to conduct these studies since the workshop happened when the academic year was starting.

Actual research output was also examined to check the details of the participant's study. The results revealed that their areas of interests are focused on online distance learning. This result shows that the research priority of the faculty and administrators is related to technology integration. Perhaps the social issues and conditions at the time the study was conducted was also contributory to this priority. There was a global pandemic; hence, face-to-face schooling was not allowed. All schools in the Philippines went to teaching online. The institution is the first school that implemented online distance learning. Consequently, faculty members and administrators saw this as the most relevant topic to study to determine the efficacy of the teaching practices. Some topics include the use of social media, online academic honesty, designing authentic online assessments, online values formation, use of learning playlist, flipped learning among others. Given all these topics, as the participants develop their ARs, they need to read existing literature on how they design their lessons and programs. This process exposes them to many ideas and practices; thus, they become knowledgeable about the content and concepts, thereby making them curriculum and instructional experts—a critical component of leadership readiness because they develop their skills in making decisions about educational issues that they themselves encounter [90,91].

*3.13. Research Question 4*

4.    How can the institution sustain the teachers desired level of research excellence?

The answers to the reflection logs and evaluation form were coded, and themes were generated to determine the factors that will sustain the level of research excellence of the participants in Table 11.

**Table 11.** Themes on sustaining the STAR program as generated from the evaluation forms.

| Code | Category | Description | Generated Theme |
| --- | --- | --- | --- |
| Research Department | Assign Research Coordinator and Consultant/s | Create digital research common where researchers and other Lasallian Partners can collaborate and engage in the conduct of research. This includes Research mentoring, Think Tank sessions, and critiquing to sustain and foster a culture of research within the institution | Research Office |
| Training | Send faculty and administrators to reputable research conferences | STAR becomes a regular training for all employees | Regular Training |
| Mentor | Regular follow through of research writers | Walkthrough or mentor interested and potential Lasallian researchers | Mentoring |
| Rewards | Give incentives, merit, or compensation | Give recognition or merit to acknowledge the contribution and/or efforts of employees | Meaningful Rewards |
| Research Repository | Online Research Journal | Create a repository of all research articles | Research Journal |
| Time | Balance time for teaching and writing | Give protected time for research writing | Balanced Time |

The following sub-questions were asked from the participants to further expound the sustainability of the program.

(1) Would you recommend the STAR program to others?

The answer of all the participants is overwhelmingly yes. Reasons indicated for their recommendation are the following: (1) Creates research-driven institution. STAR will help the school establish a culture of research; (2) Comprehensive webinar workshop. The program provides effective techniques to learn action research in the shortest possible time. (3) Promotes academic vitality. It provides an opportunity to help teachers reflect on their practices. Through the program, teachers do not only gain knowledge for their own practice but can also share for the betterment and learning of others.

(2) What drives you to conduct research?

Answers include: (1) Personal accountability. Participants are highly and intrinsically motivated to conduct or pursue action research and share results; (2) Improvement of Practices. They want to improve their own practices to contribute to the overall development of the institutions learning processes; (3) Disseminate research articles. They are interested to share and contribute new knowledge. Perhaps the participants find the idea of being able to improve one's practices, and at the same time, contribute to the purposeful knowledge base and related to being a Lasallian; hence, they are driven to conduct research.

(3) What support do you need to conduct your study?

The following answers were extracted: (1) provision of reference materials; (2) resources such as funding and more time for consultation; (3) software for gathering and analyzing data quantitative and qualitative data.

With these answers, components critical to research sustainability based on existing literature were further identified. These components are: (1) capability building; (2) productivity; (3) dissemination; (4) utilization; (5) linkages. The sustainability model is shown in Figure 8.

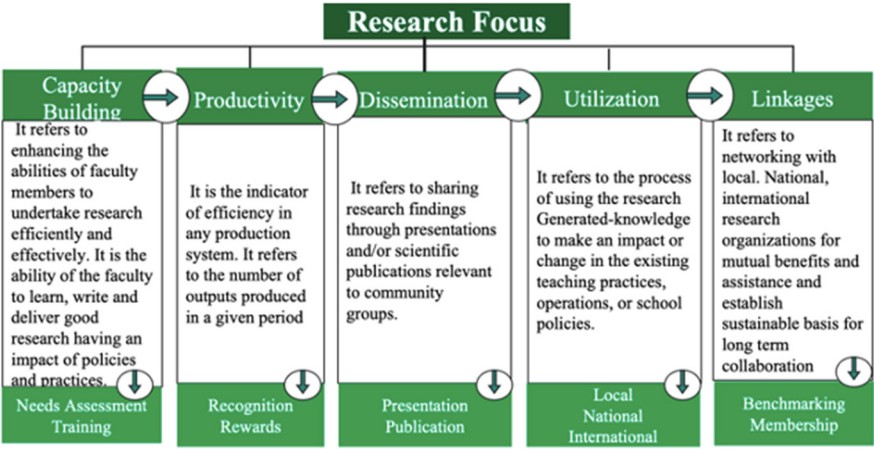

**Figure 8.** Research sustainability model.

*3.14. Research Question 6*

5. What is the effect of the use of web video conferencing on the conduct of STAR program as a model toward teacher leadership and academic vitality?

The COVID-19 pandemic compelled the Philippine educational system to enter a new era of online learning. This study investigated and provided insight into the effect of WVC on the conduct of the STAR program. To determine the effect of the video conferencing, an evaluation form was accomplished by the participants. The items were coded following the three elements of CoI. Table 12 shows the results of the evaluation.

**Table 12.** STAR program evaluation using the community of inquiry.

| I. | Social Presence (SP) | Strongly Agree | Agree | Disagree | Strongly Disagree |
|---|---|---|---|---|---|
| 1. | Web video conferencing and other technology applications were appropriate for the workshop. | 95.5 | 4.5 | 0% | 0% |
| 2. | he materials/references (handouts, meet recordings, etc.) were relevant and up to date. | 95.5 | 4.5 | 0% | 0% |
| 3. | The layout/style of the program materials/discussion helped my learning experiences. | 95.5 | 4.5 | 0% | 0% |
| 4. | There was a good mix of materials (presentations, discussion, workshop, critiquing). | 9.1 | 90.9 | 0% | 0% |
| 5. | The group size was right for the program. | 86.4 | 13.6 | 0% | 0% |
| II. | Teaching Presence (TP) | | | 0% | 0% |
| 6. | The program matched the objectives outlined in the invitation. | 100 | 0 | 0% | 0% |
| 7. | The program is relevant to the role/responsibilities that I perform. | 90.9 | 9.1 | 0% | 0% |
| 8. | The resource speakers were effective, had knowledge of the content, responded fully and I had relevant examples and inputs. | 100 | 0% | 0% | 0% |
| III. | Cognitive Presence (CP) | | | | |
| 9. | I had the opportunity to practice, apply, and gain feedback on my learning during the webinar workshop. CP/TP | 90.9 | 9.1 | 0% | 0% |
| 10. | I had the opportunity to raise questions and verify areas that concerned my understanding of the topics discussed. CP/TP | 95.5 | 4.5 | 0% | 0% |
| 11. | I will be able to apply what I have learned in my current practice. | 95.5 | 4.5 | 0% | 0% |

The answers are in full agreement to the statements. This shows that the use of WVC provided an opportunity for dynamic consultant-participants interaction and engagement, inter-participant dialogue, thereby enabling the participants to bridge the psychological and communication distance between trainers and participants and among the participants. It can also be deduced that there is a provision of the opportunity to develop the cognitive skills of analysis and autonomy.

This result is further validated through the answers to the open-ended question included to the evaluation form. These are categorized using the three elements of CoI as shown in Table 13 below.

The table above is indicative that using WVC, the three interdependent elements of CoI namely social, teaching, and cognitive presence, proved essential and successful to the synchronous and asynchronous delivery of the STAR program, thereby creating an effective educational experience for all the participants [67,92,93]. Conrad, in his literature review article, explored the impact of video technology on each of the three elements of the CoI framework. He points out that WVC helps with building a CoI framework by establishing social presence for both the instructor and the student, engaging learners, and learner-learner interactivity, increasing teaching presence by allowing robust feedback and conveying of emotions to the students, and enhancing cognitive presence by allowing a flipped classroom model of teaching, which encourages critical thinking and processing of information. The results supported those of other studies that showed that expert's presence and an interactive and collaborative style were significant determinants of participants' positive outcomes and the effectiveness of online learning environments [65,94] Additionally, the findings of the present study were consistent with Duygu's study, which revealed a successful experience of the teaching content, effectiveness, interaction, and idea-sharing of WVC [95]. Moreover, the findings were consistent with a study by Doggett that showed a positive student experience with the instructor's use of WVC technology and teaching skills. Several other studies during the COVID-19 pandemic similarly highlighted the effectiveness of WVC for delivering online teaching, engaging students, and establishing a sense of community and social presence, thereby promoting a positive experience [2,3,48,93]. The study found that training efficacy and quality relied on teaching, cognitive, and social presence and not on technology. However, technology remains an important platform that supports teachers' educational activities. Additionally, this study illuminated the importance of satisfying the growing demands for online education while maintaining a worthwhile learning experience [47]. Therefore, standards should be established for online distance training program and innovative educational methods [96–99]. Finally, this study has the potential to enhance WVC training effectiveness and enrich online training delivery in other professional development programs.

**Table 13.** STAR program evaluation.

| What Did You Find Most Beneficial about the Program? | |
|---|---|
| **CoI** | **Participant's Evaluation** |
| Teaching Presence | 1. The resource speakers are credible.<br>2. The inputs and insights of the research mentors were truly beneficial. It guided me in the conduct of my study.<br>3. The presence of highly regarded experts guides the participants in the completion of their AR.<br>4. Great support from experts<br>5. The materials and process are not just by the book but more importantly, a culmination of years of experiences of the experts.<br>6. An available support group that encourages every participant in this journey.<br>7. The workshops allowed opportunities to obtain immediate feedback and learn from what the others did and the feedback that they obtained.<br>8. It is the opportunity to present and receive comments and suggestions from the speakers.<br>9. The discussion and sharing were the most beneficial.<br>10. The sharing of tips and techniques on how to effectively conduct AR inside the classroom |

**Table 13.** *Cont.*

| CoI | | Participant's Evaluation |
|---|---|---|
| | | **What Did You Find Most Beneficial about the Program?** |
| Cognitive Presence | 1. | The presence of highly regarded experts guides the participants in the completion of their AR. |
| | 2. | The materials and process are not just by the book but more importantly, a culmination of years of experiences of the experts. |
| | 3. | An available support group that encourages every participant in this journey. |
| | 4. | Simplified frameworks for the action research matrix and the guided framework for identifying the data analysis tool to use. |
| | 5. | The step-by-step discussion of the actual use of data analysis software such as SPSS and NVIVO was also helpful and informative. |
| | 6. | The nature of the workshop program wherein we can apply what we've learned into actual research methodologies. Then, improve it based on constructive feedback of the speakers. |
| | 7. | Research presentations. |
| | 8. | The step-by-step process on how to make action research. |
| | 9. | The comprehensive explanation of the whole action research writing process. |
| Social Presence | 1. | An available support group that encourages every participant in this journey. |
| | 2. | The consultation after sync sessions. |
| | 3. | The availability of the materials and resources such as video recordings, handouts, and presentations posted in the Google Classroom. |
| | 4. | I like the nature of the program itself, the sync sessions held using video conferencing. If I cannot attend the session, I can readily view the recordings and catch up on what I missed. |
| | 5. | The knowledge of computer applications for performing statistical procedures in one's research |
| | 6. | The step-by-step discussion of the actual use of data analysis software such as SPSS and NVIVO was also helpful and informative. |
| | 7. | The friendly atmosphere and the group were mixed with different disciplines. |
| | 8. | The positive vibe and academic excellence portray in the group. |

## 4. Discussions

The disruption brought about by the COVID-19 crisis affected the entire education ecosystem. The challenges require high-level innovations and professional development programs to train teachers to become highly competent and meet the challenges of the times. The COVID-19 crisis has become an opportunity to restructure and redesign the professional development programs for educators and create new strategies for strengthening their leadership skills and academic vitality [40,47,48].

Within the professional development provided by the STAR program using WVC, teacher participants were trained to develop their leadership and academic vitality skills by being closely mentored and guided to study the processes, principles and reflect on their attitude on the conduct of actions research and integrate technology in their teaching. As espoused by the AR principles wherein self-reflection is a major component of performing AR, participants were provided an opportunity to study, analyze, and improve their own records of practices. The teachers who partook in the STAR professional development, through a supported teacher network, made changes in the way they act, do, plan, and study for their teaching, resulting in the manifestation of improved leadership skills and vitality.

The success of the STAR program, while drawing from a multitude of effective practices that were both embedded in the online course, rests upon these central factors.

1. The use of web video conferencing to deliver the program efficiently and effectively in a remote learning environment.

Through the WVC platform, the three elements of the CoI, namely teaching presence, social presence, and cognitive presence, are evidently shown in the teaching and learning process. The three presences were clearly established within the delivery of the STAR program based on the comments of the participants. The study proved that there is a relationship between the three presences and participants' learning, satisfac-

tion with the PD program, satisfaction with the consultants, actual learning, and sense of belonging [62,65,68]. With the clear establishment of the three presences, meaningful learning online has been achieved.

2. The nature of the workshop program

The program was designed in such a way that the principles and processes were discussed and applied in the actual classroom/school experiences of the participants. By allowing the teachers to focus on their own content or practice, engaging them in active learning and inquiry through the conduct of their own action research, they were able to maximize the learning opportunities and experiences provided to them. Their perceptions and actual conduct of action research significantly improved as evidenced by the various and multiple outputs of quantitative and qualitative data collected in this study. Their perceptions on technology integration, actual use of the technology tools, and their research priority areas have all shown how they view the importance of technology integration in actual teaching and learning. It is evident that the drivers of teacher leadership and academic vitality, which are the conduct of research and technology integration, have been met.

Currently, with the ongoing discussion on managing the continuity of learning and reopening of schools across the globe, it pays to have such professional programs provided for teachers because this will empower them to demonstrate leadership skills by being able to adapt and adjust to the new learning setup and contribute to the educational reforms [40]. Through the conduct of the STAR program, a research sustainability model was created. As the title suggests, this was created to ensure the continuity of the STAR program in the institution. Perhaps other institutions may use the model for their own professional development initiatives.

## 5. Conclusions

As demonstrated in this study, the use of WVC had an overall positive outcome on participants performance. Video conferencing has flourished with the development of various services on the Internet. Video conferencing, as a way of open and distance learning, provides education and training in a more flexible way than the regular teaching methods. Technology is an essential component of this training program. However, training quality relies on teaching, cognitive, and social presence, rather than on technology. Nonetheless, technology remains an important platform that supports professional development activities. The completion of this study reflected positively on the success on the use of WVC to conduct the STAR program. Thereby, affirming that with the halt of face-to-face learning, learning can be achieved.

Relatedly, there is evidence that sustaining teacher leadership and academic vitality can be achieved in an institutional level. In the present educational setup where educational reforms are necessary, it is important to give greater attention to the possibilities of creating a teacher community, which has the knowledge and expertise to solve the challenges of a more complex and diverse student body. In this respect, teachers equipped with the leadership skills and academic vitality will raise the profile and relevance of high-quality functioning educational systems.

STAR promotes teacher leadership and academic vitality. The key drivers are action research, leadership, and technology integration. This program hones the potentials of faculty members and administrators to be able to deliver quality instructions, and at the same time, become research oriented and integrate technology effectively in teaching and learning. As they become research practitioners, they become academically vital, and their leadership skills are developed.

Existing literature proves that building teachers' capacity to conduct research develops not only the ability to write action research, but also, they develop many other essential skills. When they reflect on their own teaching experiences using the lens of action research, they become more critical of their own practices, thus challenging the status quo. With this, they start to find ways on how they address the issues they encounter by designing lessons

and activities that are geared toward the improvement of their current practices. Since the goal of action research is about improving one's own practices, the process of performing action research itself is already making the teachers become academically vital.

Putting them all together, it is evident that the STAR program promotes teacher leadership and academic vitality as represented by the model in Figure 9.

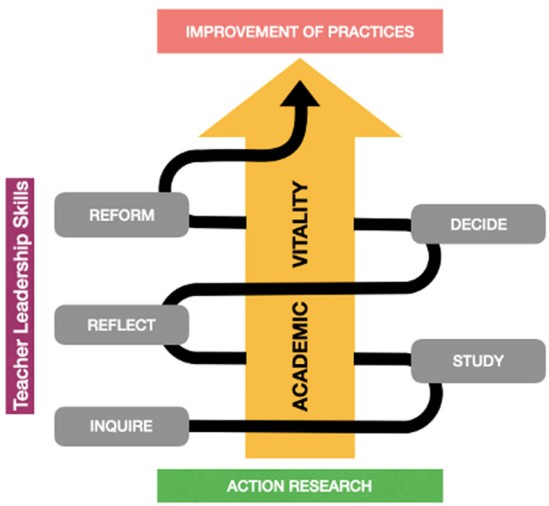

**Figure 9.** Teacher leadership and academic vitality model.

The conduct of action research leads the faculty and administrators to perform an in-depth review and study of their teaching practices. In the context of the institution where the study was conducted, technology integration is subsumed in the actual action research studies of the participants. As they discover the answers to their queries, they can improve learning and transform instructions, making effective educational reforms. This helps them develop leadership skills because they make crucial decisions about their own practices, and they become critical to the challenges to the institution.

Moving forward, teacher leadership and the academic vitality model can be extended to other institutions who intend to develop the leadership readiness and academic vitality of their own faculty and administrators. They may replicate the STAR program following the use of WVC and can customize the content of their professional program based on the goals and priorities of their own institutions.

**Author Contributions:** Conceptualization, J.B.T., M.S.P. and S.E.A.; methodology, J.B.T.; software, J.B.T., M.S.P. and S.E.A.; validation, M.S.P. and S.E.A.; formal analysis, J.B.T.; investigation, J.B.T.; resources, J.B.T.; data curation, J.B.T.; writing—original draft preparation, J.B.T.; writing—review and editing, M.S.P. and S.E.A.; visualization, J.B.T.; supervision, M.S.P. and S.E.A.; project administration, J.B.T., M.S.P. and S.E.A. All authors have read and agreed to the published version of the manuscript.

**Funding:** This research received no external funding.

**Institutional Review Board Statement:** Not applicable.

**Informed Consent Statement:** Not applicable.

**Data Availability Statement:** Not applicable.

**Acknowledgments:** The authors acknowledge support from De La Salle Zobel School.

**Conflicts of Interest:** The authors declare no conflict of interest.

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
