# Peer review of "Using Web Video Conferencing to Conduct a Program as a Proposed Model toward Teacher Leadership and Academic Vitality in the Philippines"

_education, doi:10.3390/educsci11110658_

Round 1
Reviewer 1 Report
The article Using Web Video Conferencing to Conduct the STAR (Sustaining Teacher-Leadership and Academic-Vitality through Research) Program as a Proposed Model towards Teacher Leadership and Academic-Vitality is an interesting and actual mix study that aimed to evaluate the effect of using Web Video Conferencing to conduct the STAR (Sustaining 10 Teacher-Leadership and Academic-Vitality through Research). I think so that it is an excellent and necessary article. However, I believe that some aspects could be improved or clarified, which is why I am making some suggestions and requests for clarification. In addition, I think it is necessary that many parts of the text are summarised and a specific discussion section should be included.
Abstract and Key Words
Please remove acronyms from the abstract, or at least indicate their meaning in brackets. The reader may have difficulty understanding the article on a first reading.
Reduce the number of key words between 3 and 5. Avoid compound terms that make them difficult to search in the database.
Title
The title is, in my opinion, too long. Perhaps it could be shortened as follows or something similar: Using Web Video Conferencing to Conduct a Program as a Proposed Model towards Teacher Leadership and Academic-Vitality.
It would also be useful to indicate the geographical location where the research has been carried out.
Introduction
- It’s clear that the use of WBC has increased markedly as a consequence of the COVID-19 pandemic. Although it is commented on, it would be good for the reader to be able to know data in this regard. I think this is a central element in understanding the importance of the issue.
- The introduction is comprehensive, and allows the reader to situate himself on the issue. However, I find it too long for an academic article. It would be desirable to summarise certain points more, and to eliminate certain headings, such as definition term, which could be included in an annex.
Method and Approach
The study presents a robust and appropriate methodology. I would venture to say excessively detailed, although this is not a problem. I think it is a good idea to use a mixed research method in this case. However, there are some issues I need to clarify:
- You indicate that you have followed the thematic approach for qualitative research. However, it is necessary to indicate what they are referring to specifically, and to indicate the reference bibliography in this respect.
- You indicate that the questionnaire used is the Prudente and Aguja questionnaire. I understand that this is a validated questionnaire for the Philippine context?
- Has authorisation been obtained from any organisation's ethics committee, beyond the permission of the school president?
- As with the introduction, the methodology is too long. The Star Logo, for example, is not necessary at all. The same goes for some other things. Many things can be summarised to make the paper easier to read.
Results
- As with other sections, the results are adequate, but again it is possible to summarise them. For example, the hastags indicated do not contribute anything to the reader.
- ed should not be explained, which is already done in the methodology, but the results obtained are sufficient.
- Tables 4 and 9 are difficult to read. You need to summarise it and make it easier to read.
Discussion
The text lacks a discussion section as such. Although the author(s) do so throughout the results, especially in the last part, this should be a separate and more extensive heading. It is essential for the research to confront the results with other experiences and with what has happened in other contexts, especially with the revolution brought about by the COVID-19 pandemic confinements.. It is essential, therefore, to include this section.
Conclusions
The conclusions are adequate in length and content
Finally, I congratulate you on the study and encourage you to continue studying in this line.

Reviewer 2 Report
This paper aimed to evaluate the effectiveness of Web Video Conferencing as an option for faculty members' professional development and leadership and academic vitality using CIF.
The authors have produced a report with depth in the research area and effectively utilized a mixed method to research the problem.
The structure and quality of presentation is commendable.
The researcher may expand the criteria for the selection of participants for this study, they have only mentioned the "willingness to participate in a nine-week program", what were the other selection metrics for arriving at the suitability of the 33 participants.
Could this report be written as two separate paper?
Round 2
Reviewer 1 Report
Congratulations on the changes introduced. Please reduce the number of key words and improve them. It is essential for a good position in databases.